# Role of Selective Digestive Decontamination in the Prevention of Ventilator-Associated Pneumonia in COVID-19 Patients: A Pre-Post Observational Study

**DOI:** 10.3390/jcm12041432

**Published:** 2023-02-10

**Authors:** Emanuela Biagioni, Elena Ferrari, Ilenia Gatto, Lucia Serio, Carlotta Farinelli, Irene Coloretti, Marta Talamonti, Martina Tosi, Marianna Meschiari, Roberto Tonelli, Claudia Venturelli, Cristina Mussini, Enrico Clini, Mario Sarti, Andrea Cossarizza, Stefano Busani, Massimo Girardis

**Affiliations:** 1Intensive Care Unit, University Hospital of Modena, 41124 Modena, Italy; 2Infectious Disease Unit, University Hospital of Modena, 41124 Modena, Italy; 3Respiratory Disease Unit, University Hospital of Modena, 41124 Modena, Italy; 4Virology and Molecular Microbiology, University Hospital of Modena, 41124 Modena, Italy; 5Immunology Laboratory, University of Modena and Reggio Emilia, 41124 Modena, Italy

**Keywords:** COVID-19, acute respiratory distress syndrome, intensive care unit, mechanical ventilation, treatment

## Abstract

The aim of our study was to evaluate whether the introduction of SDD in a structured protocol for VAP prevention was effective in reducing the occurrence of ventilator-associated pneumonia (VAP) in COVID-19 patients without changes in the microbiological pattern of antibiotic resistance. This observational pre-post study included adult patients requiring invasive mechanical ventilation (IMV) for severe respiratory failure related to SARS-CoV-2 admitted in three COVID-19 intensive care units (ICUs) in an Italian hospital from 22 February 2020 to 8 March 2022. Selective digestive decontamination (SDD) was introduced from the end of April 2021 in the structured protocol for VAP prevention. The SDD consisted of a tobramycin sulfate, colistin sulfate, and amphotericin B suspension applied in the patient’s oropharynx and the stomach via a nasogastric tube. Three-hundred-and-forty-eight patients were included in the study. In the 86 patients (32.9%) who received SDD, the occurrence of VAP decreased by 7.7% (*p* = 0.192) compared to the patients who did not receive SDD. The onset time of VAP, the occurrence of multidrug-resistant microorganisms AP, the length of invasive mechanical ventilation, and hospital mortality were similar in the patients who received and who did not receive SDD. The multivariate analysis adjusted for confounders showed that the use of SDD reduces the occurrence of VAP (HR 0.536, CI 0.338–0.851; *p* = 0.017). Our pre-post observational study indicates that the use of SDD in a structured protocol for VAP prevention seems to reduce the occurrence of VAP without changes in the incidence of multidrug-resistant bacteria in COVID-19 patients.

## 1. Introduction

The occurrence of bacterial pneumonia, mostly ventilator-associated pneumonia (VAP), has been reported as a frequent complication in critically ill COVID-19 with an incidence ranging between 40–60%, which is roughly three times larger than in non-COVID-19 patients [1,2,3]. This high incidence has been attributed to several factors, such as the scarce application of the standard infection control rules, the high use of pronation and myorelaxants, and the host immune dysfunction caused directly by the SARS-CoV2 infection and by the use of steroids and immunosuppressive agents [4,5]. Similarly to other patients admitted to the intensive care unit (ICU), in COVID-19 patients, the development of secondary infections prolongs the length of mechanical ventilation and hospitalization, increasing the mortality risk [6]. 

As experienced in many other centers, even in the three ICUs dedicated to the COVID-19 patients of our hospital, we observed a dramatic increase in the incidence of VAP, moving from 8–10% in non-COVID-19 mechanically ventilated patients in 2019 to 35–40% during the first six months of SARS-CoV2 pandemic. Unfortunately, although specific interventions for reinforcing compliance with the infection control rules and the interventions were included in the internal protocol for VAP prevention [7], the incidence of VAP in COVID-19 patients also remained very high in the following six months of the pandemic. Therefore, in April 2021, we decided to introduce in our protocol the use of selective digestive tract decontamination (SDD). The use of SDD for reducing the VAP was proposed more than 30 years ago [7,8] but it is still sporadically used worldwide even though several trials indicated reasonable evidence of benefit without detrimental effects on bacterial resistance selection [9,10]. In COVID-19 patients, some studies investigated the effects of SDD in preventing secondary infections with promising results [11,12,13]. 

This observational pre-post study aimed to evaluate whether the introduction of SDD in a structured protocol for VAP prevention was effective in reducing the occurrence of VAP in COVID-19 patients without changes in the microbiological pattern of antibiotic resistance.

## 2. Materials and Methods

In this observational study, we included all the adult patients requiring invasive mechanical ventilation (MV) for severe respiratory failure related to SARS-CoV-2 infection admitted to the three COVID-19 ICUs of the University Hospital of Modena from February 2020 to March 2022. Patients with an ICU length of stay (LOS) < 24 h and limitation of care or do not resuscitate orders were excluded from the study. The Institutional Ethics Committee of Area Vasta Emilia Nord (EC AVEN) approved the study (approval number 396/2020/OSS/AOUMO—CoV-2 MO-Study). Due to the study’s observational nature, written informed consent was not required.

### 2.1. Treatment Protocol

All the patients received standard ICU supportive care and specific COVID-19 therapies according to WHO guidelines [14] and national protocols [15] for treating COVID-19 patients. In addition, the local protocol allowed the use of methylprednisolone at 2 mg/kg/day to prevent the onset of pulmonary fibrosis in patients with acute respiratory distress syndrome (ARDS) who maintained at a PaO_2_/FiO_2_ ratio < 150 mmHg for at least seven days of MV [16] and Tocilizumab (TOCI) in patients with moderate or severe ARDS since March 2020. The internal protocol for VAP prevention in use before the pandemic (supplemental material) was routinely applied in all COVID-19 patients with invasive MV. The protocol also included using nasal mupirocin for five days and chlorhexidine for body cleaning at least once daily. However, the high incidence of VAP observed in the first six months of the pandemic (February–April and September–November 2020) led to an urgent multifaceted program (including audit and educational meetings (by electronic platforms) and practical simulation) for reinforcing compliance with the internal protocol. Due to the persistently high rate of VAP observed in the three months following the multifaceted program, from the end of April 2021, we decided to also introduce SDD into the protocol. The SDD consisted of a tobramycin sulfate, colistin sulfate, and amphotericin B suspension that was applied in the patient’s oropharynx and the stomach via a nasogastric tube four times per day for ten consecutive days or until endotracheal tube removal. 

### 2.2. Data Collection

Patient demographics, Simplified Acute Physiology Score II (SAPS II), and standard laboratory test results, including the coagulation and inflammatory variables, were collected at ICU admission. In addition, therapy with steroids, tocilizumab (also before ICU admission), the occurrence of the first bacterial VAP, early onset VAP (EVAP), late-onset VAP (LVAP), the onset time from the start of invasive MV to VAP, the microorganisms causing VAP and their resistance pattern, CMV blood reactivation, and probable invasive pulmonary aspergillosis were collected during the ICU stay. The length of invasive MV and hospital stay and hospital mortality were also recorded. 

As for the ICU protocol before the pandemic, the patients were microbiologically screened at ICU admission and twice a week with the rectal swab and tracheal aspirate, serum Galactomannan, and quantitative cytomegalovirus DNAemia in the blood. Further microbiological examinations were performed if there was clinical suspicion of infection. According to international guidelines [17,18], the occurrence of bacterial VAP was defined as the presence of a new persistent infiltrate observed at the chest radiograph or computed tomography scan at least 48 h after orotracheal intubation associated (at least one) with the worsening of oxygenation, purulent bronchial secretions, leukocytosis, and fever, and the presence of potentially pathogenic microorganisms in culture from tracheal aspirate and bronchoalveolar lavage. The clinical and microbiological data have been controlled and revised by an infectious disease specialist (MM) and a well-experienced intensivist (BM). Bacterial VAP was defined as EVAP if occurring within 96 h from invasive MV initiation or LVAP if occurring later. Multidrug-resistant (MDR) microorganisms were defined if the isolate was non-susceptible to at least one agent in three antimicrobial categories listed in the standard definitions for acquired resistance [19]. The CMV reactivation was set for a DNAemia >62 UI/mL in the whole blood, the detection threshold of the method used. Probable invasive pulmonary aspergillosis was defined according to definitions from the recent consensus document [20].

### 2.3. Data Analysis

We used a Cox proportional hazards regression model, including variables with *p*-value < 0.1 at unadjusted analysis to evaluate the independent association of SDD therapy with VAP occurrence censored at day 60. We also performed a secondary analysis by matching patients with and without SDD use (1:1) using a propensity score estimated by a multivariable logistic-regression model that included as covariates the risk factors to be treated with SDD; the nearest-neighbor method was applied to the propensity-score matching analysis.

Non-parametric and χ^2^ tests were used as appropriate for comparing demographic and baseline values and outcomes in patients with and without SDD and VAP. All results were expressed as medians (range) for continuous variables and as frequencies (percentage) for categorical variables. All tests were two-tailed, with a *p*-value < 0.05 considered significant. SPSS version 22.0 package (SPSS Inc., Chicago, IL, USA) was used to perform statistical analysis.

## 3. Results

In the study period, 591 patients were admitted to the three COVID-19 ICUs, of whom 348 fulfilled the inclusion criteria, and 86 (32.8%) received SDD during their ICU stay. At ICU admission, the patients who received SDD were younger, with lower SAPS II scores, PaO_2_/FiO_2_ values, and medical history of hypertension than patients who did not receive SDD. In addition, all these patients received steroids compared to around 90% of the patients in the no-SDD group (Table 1). 

Bacterial VAP occurred in about 30% of patients with no substantial differences throughout the waves (Appendix A) with an onset median time of 8 (IQR 5–14) days after the initiation of invasive mechanical ventilation (Figure 1). 

In patients who received SDD, we observed a 7.7% absolute risk reduction (*p* = 0.192) for VAP compared to those who did not receive SDD (Table 2). This reduction was mainly sustained by a reduction in the occurrence of late VAP. Gram-negative microorganisms were isolated more frequently than Gram-positive, with some differences in species between the two groups (*p* = 0.259). In the patients who received SDD, we observed a lower rate of *Pseudomonas aeruginosa* and *Klebsiella* spp. and a higher rate of *Escherichia coli* and *Serratia marcescens* compared to patients who did not receive SDD. The onset time of VAP, the occurrence of multidrug-resistant patterns in microorganisms causing VAP, CMV blood reactivation, and probable invasive pulmonary aspergillosis were similar between the two groups. Similarly, the length of invasive MV and hospital stay and hospital mortality did not differ between patients who received and who did not receive SDD (Table 2).

Univariate analysis showed that SDD reduced VAP occurrence (HR 0.590, CI 0.373–0.933; *p* = 0.024) and adjusted analysis for confounders confirmed this association (HR 0.536, CI 0.338–0.851; *p* = 0.017) (Table 3). 

The secondary analysis of the 162 patients matched (1:1) for the individual propensity to receive SDD showed that the use of SDD provided an 11% absolute risk reduction in VAP development (*p* = 0.133) without changes in antibiotic resistance pattern. This reduction was not associated with reducing invasive MV length and hospital mortality. The 82 pairs of patients matched by propensity score were well balanced for demographics, comorbidities, and characteristics at ICU admission (Appendix A). 

## 4. Discussion

This observational pre-post study confirmed that VAP occurs in around a third of COVID-19 patients requiring invasive mechanical ventilation and that in these patients, the use of SDD integrated into a structured protocol for VAP prevention may reduce the VAP occurrence, especially late VAP, without changes in microorganism patterns of resistance to antibiotics. 

The high incidence of VAP observed in our COVID-19 patients was consistent with data reported in other studies [1,2]. Interestingly, the incidence of VAP did not change during the study period, supporting the hypothesis that the scarce application of infection control recommendations, which in our center could have occurred during the first surge from February to April 2020, explains only partially the high risk of VAP in COVID-19 patients compared to no-COVID-19. Moreover, as indicated by others, the development of VAP was related to increased hospital stay and mortality rates. 

The role of SDD in preventing low respiratory tract infection during invasive MV has been extensively investigated over the last 15 years by more than 40 interventional trials enrolling more than ten-thousand patients [7,21,22]. According to data, with an illustrative incidence of low respiratory tract infection between 30% to 40 %, SDD may provide a low to moderate grade of evidence of a reduction for low respiratory tract infection ranging from 24% when combined with systemic antibiotics to 14% without systemic antibiotics [21]. In our population, the benefit provided by SDD was lower than that observed in COVID-19 patients but quite similar to that reported in no-COVID-19 patients without systemic antibiotics. However, in COVID-19, two observational studies showed a reduction in VAP occurrence from about 55% in the control patients (*n* =470) to 25% in patients (*n* =170) treated with SDD and systemic antibiotics [12,13]. The lower incidence of VAP in our control group (34%) compared to the control group (55%) of the two observational trials [12,13] on COVID-19 and the use of systemic antibiotics may explain the difference observed. In our cohort, the reduction in VAP incidence was sustained mainly by a reduction in late VAP. Previous data demonstrated that SDD could reduce both early and late VAP (17–18). This difference could be due to the low rate of early VAP in our population, probably caused by the little or no necessity of emergency intubation in COVID-19 patients compared to other populations, such as trauma or cardiac arrest patients. As concerns the risk of the development of antibiotic resistance, according to other studies [23,24], we did not observe any change in the resistance pattern of microorganisms isolated in the respiratory tract and other sites, including rectal colonization. Finally, despite the reduction in VAP, the need for invasive mechanical ventilation, hospital stay, and hospital mortality did not change using SDD. The same results have been observed in several trials, mainly when SDD was used without the association of intravenous antibiotics [7,21]. In COVID-19 patients, the study by Massart et al. [12] using a multisite decontamination strategy including SDD and intravenous antibiotics observed a 14% reduction in hospital mortality, similar to the reduction in 28-day mortality observed by Luque-Paz et al. [13]. This discrepancy with the results observed in our cohort could be attributed to differences in the VAP incidence in the control groups (see above), the use of systemic antibiotics, and the population characteristics.

Compared with previous studies of SDD in COVID-19, the strength of our study is the evaluation of the net effect of adding SDD into a structured protocol for VAP prevention without the interference of other preventive interventions (e.g., systemic antibiotics, multisite decontamination, reinforcement of standard procedures). However, the study has significant limitations. Firstly, the study design and the sample size limit the general applicability of the results observed. For instance, the difference in PaO_2_/FiO_2_ between the patients receiving and not receiving SDD may indicate non-protocolized changes in ICU admission criteria among the different COVID-19 surges. Second, the VAP incidence may be underestimated because, due to the difficulties in clinical diagnosis of secondary pneumonia in COVID-19 patients, only the microbiologically proven VAP has been considered. However, this could have generated minimal bias in comparing groups because the same method has been used in controls and patients with SDD. Third, the lack of systematic screening for MDR bacteria colonization limits the results observed on the effects of SDD use on the selection of MDR strains. Finally, our study showed an increased risk of VAP related to using tocilizumab but not steroids, as demonstrated by others [5,6]. However, it was beyond the aim of our study to further investigate this point.

## 5. Conclusions

In conclusion, according to other experiences, our pre-post observational study in COVID-19 patients suggests an association between the use of SDD in a structured protocol for VAP prevention and the reduction in VAP occurrence, especially late VAP, without any increase in the incidence of VAP sustained from multidrug-resistant bacteria. However, this decrease was not associated with any benefit in survival rate. Due to the study’s limitations, the results should be considered only as generating hypotheses for future trials.

## Figures and Tables

**Figure 1 jcm-12-01432-f001:**
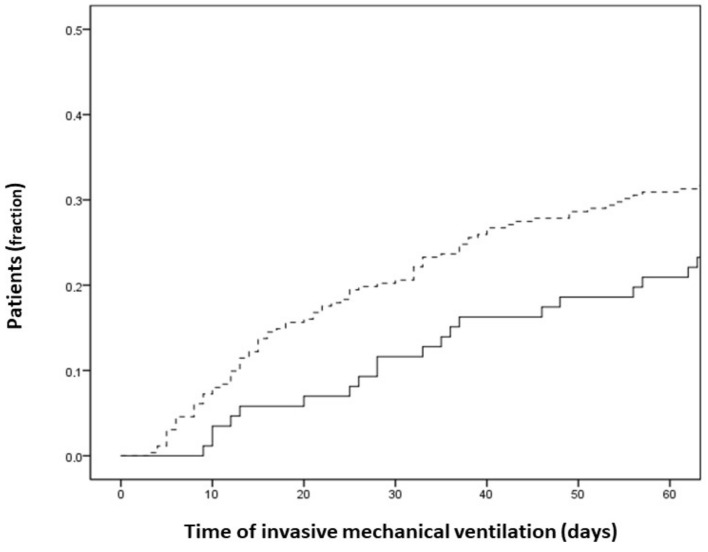
Cumulative probability of developing ventilator-associated pneumonia after initiation of invasive mechanical ventilation in patients receiving (solid line) and non-receiving (dotted line) selective digestive decontamination.

**Table 1 jcm-12-01432-t001:** Demographics, comorbidities, severity scores, and laboratory at ICU admission in all patients and in patients with or without selective digestive decontamination. The use of steroids during ICU stay and of tocilizumab before and during ICU stay are also reported.

Patients Characteristics and Laboratory at ICU Admission	All Patients (*n* = 348)	No SDD (*n* = 262)	SDD (*n* = 86)	*p*-Value
Age (years; median, IQR)	67 (60–73)	67 (61–74)	64 (57–71)	0.037
Sex, male (*n*, %)	264 (75.9)	205 (78.2)	59 (68.6)	0.070
BMI (kg/m^2^; median, IQR)	29 (26–33)	29 (26–33)	31 (26–33)	0.153
Comorbidities				
Diabetes (*n*, %)	81 (23.3)	60 (22.9)	21 (24.4)	0.773
Hypertension (*n*, %)	177 (50.9)	145 (55.3)	32 (37.2)	0.004
Immunosuppression (*n*, %)	62 (17.8)	46 (17.6)	16 (18.6)	0.826
SAPS II (median, IQR)	35 (32–42)	36 (32–42)	33 (29–42)	0.007
D-dimer (ng/mL; median, IQR)	1645 (905–3820)	1655 (910–3260)	1638 (899–6686)	0.462
Lymphocyte count (10^9^/L; median, IQR)	0.62 (0.43–0.90)	0.63 (0.44–0.89)	0.62 (0.42–0.98)	0.878
Platelet count (10^9^/L; median, IQR)	207 (159–282)	202 (156–274)	220 (169–290)	0.230
PaO_2_/FiO_2_ (mmHg; median, IQR)	95 (75–123)	98 (78–128)	88 (70–109)	0.007
Steroids (*n*, %)	319 (91.9)	233 (89.3)	86 (100.0)	0.002
Tocilizumab (*n*, %)	285 (81.9)	210 (80.2)	75 (87.2)	0.140

SDD: selective digestive decontamination; BMI: Body Mass Index; SAPSII: Simplified Acute Physiology Score II.

**Table 2 jcm-12-01432-t002:** Ventilator-associated pneumonia, and other infections during intensive care stay and hospital mortality in all the patients and in patients receiving and not receiving selective digestive decontamination.

Infections during ICU Stay	All Patients (*n* = 348)	No SDD (*n* = 262)	SDD (*n* = 86)	*p* Value
VAP (*n*, %)	113 (32.5)	90 (34.4)	23 (26.7)	0.191
Early VAP (*n*, %)	26 (7.5)	20 (7.6)	6 (7.0)	0.841
Late VAP (*n*, %)	87 (25.0)	70 (26.7)	17 (19.8)	0.197
VAP Onset time after invasive MV (days; median, IQR)	8 (5–14)	8 (5–14)	7 (5–18)	0.766
VAP Gram-stained microorganisms				0.688
Gram-positive (*n*, %)	36 (31.3)	28 (30.4)	8 (34.8)	
Gram-negative (*n*, %)	79 (68.7)	64 (69.6)	15 (65.2)	
VAP microorganisms species (*n*, %)				0.259
*Staphylococcus aureus*	31 (27.4)	24 (26.7)	7 (30.4)	
*Pseudomonas aeruginosa*	29 (25.7)	26 (28.9)	3 (13.0)	
*Klebsiella* spp.	22 (19.5)	19 (21.1)	3 (13.0)	
*Escherichia coli*	5 (4.4)	3 (3.3)	2 (8.7)	
*Serratia marcescens*	9 (8.0)	6 (6.7)	3 (13.0)	
Other *	17 (15.0)	12 (13.3)	5 (21.7)	
VAP MDR microorganisms (*n*, %)	48 (42.5)	41 (45.6)	7 (30.4)	0.190
CMV blood reactivation (*n*, %)	107 (31.0)	78 (30.1)	29 (33.7)	0.531
Probable Invasive Pulmonary Aspergillosis (*n*, %)	82 (23.9)	59 (22.5)	23 (28.4)	0.279
Invasive MV length (days; median, IQR)	9 (5–25)	9 (5–22)	10 (5–33)	0.300
Hospital length of stay (days; median, IQR)	26 (17–42)	26 (18–40)	29 (17–43)	0.430
Hospital mortality (*n*, %)	174 (51.2)	131 (50.4)	43 (53.8)	0.598

SDD: selective digestive decontamination; VAP: ventilator-associated pneumonia; MDR: multidrug-resistant; CMV: cytomegalovirus; MV: mechanical ventilation; * Morganella Morgani, Enterobacter Cloacae, Citrobacter K, Proteus spp., Stenotrophomonas M.

**Table 3 jcm-12-01432-t003:** Demographics, comorbidities, severity scores and laboratory at ICU admission, use of steroids and tocilizumab, hospital length of stay, and mortality in patients with or without ventilator-associated pneumonia. Unadjusted and adjusted analyses (see methods) are also reported.

Patients Characteristics, Therapies, and Mortality	No VAP (*n* = 253)	VAP (*n* = 113)	Unadjusted HR (95% CI); *p*-Value	Adjusted HR (95% CI); *p*-Value
Age (years; median, IQR)	66 (58–73)	68 (61–72)	1.002 (0.984–1.022); 0.800	
Sex, male (*n*, %)	175 (74.5)	89 (78.8)	0.695 (0.442–1.092); 0.115	
BMI (kg/m^2^; median, IQR)	29 (26–33)	30 (26–34)	0.993 (0.964–1.023); 0.656	
Comorbidities				
Diabetes (*n*, %)	55 (23.4)	26 (23.0)	1.124 (0.725–1.743); 0.601	
Hypertension (*n*, %)	113 (48.1)	64 (56.6)	0.808 (0.557–1.173); 0.263	
Immunosuppression (*n*, %)	42 (17.9)	20 (17.7)	0.767 (0.472–1.244); 0.282	
SAPSII (median, IQR)	35 (30–43)	35 (33–40)	0.989 (0.969–1.008); 0.256	
D-dimer (ng/mL; median, IQR)	1660 (931–3260)	1640 (800–5190)	1.000 (1.000–1.000); 0.020	1.000 (1.000–1.000); 0.012
Lymphocyte count (10^9^/L; median, IQR)	0.64 (0.46–0.90)	0.58 (0.40–0.91)	0.944 (0.821–1.085); 0.415	
Platelet count (10^9^/L; median, IQR)	219 (163–282)	195 (148–274)	1.001 (0.999–1.003); 0.547	
PaO_2_/FiO_2_ (mmHg; median, IQR)	92 (70–123)	97 (83–122)	0.999 (0.995–1.003); 0.733	
Selective Digestive Decontamination (*n*, %)	63 (26.8)	23 (20.4)	0.590 (0.373–0.933); 0.024	0.536 (0.338–0.851) 0.008
Steroids (*n*, %)	208 (88.5)	111 (99.1)	3.658 (0.509–26.305); 0.198	
Tocilizumab (*n*, %)	185 (78.7)	100 (88.5)	1.887 (1.059–3.365); 0.031	2.034 (1.137–3.637) 0.017
IMV length (days; median, IQR)	6 (4–14)	25 (12–42)	1.003 (0.996–1.011); 0.416	
Hospital length of stay (days; median, IQR)	23 (16–35)	38 (25–59)	1.002 (0.995–1.009); 0.626	
Hospital mortality (*n*, %)	108 (46.8)	68 (61.3)	0.950 (0.646–1.396); 0.793	

SDD: selective digestive decontamination; BMI: body mass index; SAPSII: simplified acute physiology score II; VAP: ventilator-associated pneumonia.

## Data Availability

The data presented in this study are available on request from the corresponding author. The data are not publicly available due to patients’ privacy.

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
