# Peer review of "Role of Selective Digestive Decontamination in the Prevention of Ventilator-Associated Pneumonia in COVID-19 Patients: A Pre-Post Observational Study"

_jcm, 2023, doi:10.3390/jcm12041432_

Round 1

Reviewer 1 Report

Biagioni Emmanuela et al. submitted an original article entitled "role of selective digestive decontamination in the prevention of ventilator-associated pneumonia in covid-19 patients: a pre-post observational study". The authors report a reduction of VAP incidence after implementation of SDD in their protocols for preventing VAP in ICU. 

This is an interesting study adding to the debate on SDD that has been stimulated recently by two major papers in JAMA. The main interest here is the targeted population (Covid-19) and the use of this procedure in an environment with a high incidence of MDR. Detractors of SSD, which are many, often highlight the potential effects of SDD on ecology and the fact that most studies have been conducted in the Netherlands where incidence of MDR is low. For all these reasons, this study is important in this field. 

Few comments

1. Meta-analysis mainly showed that SDD reduces mortality when an intravenous antibiotic is administered for the first 2 to 4 days (Roquilly et al. Clin Infect Dis 2015).  Here I did not find the antimicrobial stewardship protocol. It seems that only the topical antibiotics (aimed to protect against late-onset VAP) were given. However, a large proportion of patients may have received IV antibiotics at ICU admission. We need to have these data to determine if in real life the patients received only topical SDD or both topical and IV. Most of time, even if IV antibiotics are not in the SDD protocol, a large proportion of patients received them (in most studies, more than 70%, which suggests that, with or without SDD, there is an overuse of antibiotics). 

2. The aim of the study is to assess the rate of VAP. However, SDD, due to local effect, may negative the results of bronchial samples. Thus we need to know the criteria for VAP diagnosis in routine, if there was a "senior review" for each diagnosis, even retrospectively. In addition, I would like to know the trend for positive blood culture and other infections during the study. 

3. There is a definition of MDR bacteria in the method but I cannot find this in the result section. It would be of interest to determine the rate of colonization, if feasible, of the patients with and without SDD. 

4. Your study shows association; you should change your sentence adding "suggests an association between A and B" (conclusion)

Minor

1. Most patients received suppressive drugs. This could have affected (1) the rate of VAP; (2) the effects of SDD. Could you comment on this?

2. I would be interested by following the inflammatory markers in the patients with and without SDD, including the rate of secondary CMV reactivation. In the same line, if you have the CT for SARS-CoV-2 in the two groups, that can be of interest.

3. Rewrite as: Pseudomonas aeruginosa; Escherichia coli; Serratia marcescens

4. Fig. 1 I do not understand why you did not include the time of invasive MV in the two groups?

5. I suggest adding the two recent studies (RCT and SR) published in JAMA on SDD in the reference list

Author Response

Reviewer 1.

  1. Meta-analysis mainly showed that SDD reduces mortality when an intravenous antibiotic is administered for the first 2 to 4 days (Roquilly et al. Clin Infect Dis 2015).  Here I did not find the antimicrobial stewardship protocol. It seems that only the topical antibiotics (aimed to protect against late-onset VAP) were given. However, a large proportion of patients may have received IV antibiotics at ICU admission. We need to have these data to determine if in real life the patients received only topical SDD or both topical and IV. Most of time, even if IV antibiotics are not in the SDD protocol, a large proportion of patients received them (in most studies, more than 70%, which suggests that, with or without SDD, there is an overuse of antibiotics).

We thank the reviewer for highlighting this important point. Differently from other ICU populations, in COVID-19 patients, the use of antibiotics was rare before secondary infections occurred in the ICU. Our cohort's rate of patients receiving intravenous antibiotics at SDD initiation was < 10% (5/81). We added this point in the results section.

  1. The aim of the study is to assess the rate of VAP. However, SDD, due to local effect, may negative the results of bronchial samples. Thus we need to know the criteria for VAP diagnosis in routine, if there was a "senior review" for each diagnosis, even retrospectively. In addition, I would like to know the trend for positive blood culture and other infections during the study.

During the study period, the criteria and the procedures used for VAP diagnosis did not change. Before and during COVID19 our standard ICU-acquired infection procedures included a 'senior review' from a well-trained intensivist and an infectious disease specialist. This procedure is part of a multifaced in-hospital antimicrobial stewardship program (Meschiari et al. Antimicrob Resist Infect Control 2021 Aug 19;10(1):123; Meschiari et al. Antibiotics (Basel). 2022 Jun 20;11(6):826) This point was described in the methods (page 5 line 120)

For infection trend, we added BSI in the Table S1

Table 1S.

Period

Date

Patients included

VAP (n; %)

Blood Stream Infection (n; %)

First wave

February-June 2020

80

27 (33,7)

12 (15,0)

Second Wave

September 2020- February 2021

104

39 (37,5)

22 (21,1)

Third Wave

March -July 2021

93

27 (29,0)

20 (21.5)

Fourth wave

October 2021-March 2022

71

20 (28,2)

17 (23,9)

  1. There is a definition of MDR bacteria in the method but I cannot find this in the result section. It would be of interest to determine the rate of colonization, if feasible, of the patients with and without SDD.

The percentage of patients with VAP sustained from MDR is reported in table 2. Unfortunately, we have no collected data on colonization.

  1. Your study shows association; you should change your sentence adding "suggests an association between A and B" (conclusion)

We modified the conclusions as suggested.

Minor

  1. Most patients received suppressive drugs. This could have affected (1) the rate of VAP; (2) the effects of SDD. Could you comment on this?

As indicated in the study's limitations, using Tocilizumab seems to increase the risk of VAP. Still, unfortunately, the high percentage of patients receiving immunosuppressive drugs did not allow us to analyze the relationship between VAP occurrence and these drugs and the interaction of SDD with this relationship correctly.

  1. I would be interested by following the inflammatory markers in the patients with and without SDD, including the rate of secondary CMV reactivation. In the same line, if you have the CT for SARS-CoV-2 in the two groups, that can be of interest.

This is an interesting point. Unfortunately, we did not systematically collect C-Reactive protein or Ferritin and the CT of SARS-COV2 during SDD use. For CMV reactivation, we reported the data in table 2.

  1. Rewrite as: Pseudomonas aeruginosaEscherichia coliSerratia marcescens

We modified it as indicated

  1. Fig. 1 I do not understand why you did not include the time of invasive MV in the two groups?

We modified the figure with the cumulative risk for the 2 groups

  1. I suggest adding the two recent studies (RCT and SR) published in JAMA on SDD in the reference list

We added the 2 references as indicated

Reviewer 2 Report

Dear editor,

I read with great interest the manuscript entitled ‘role of selective digestive decontamination in the prevention of ventilator-associated pneumonia in covid-19 patients: a pre-post observational study’.

Prevention of VAP in the setting of COVID-19 is an important topic. This study reports a reduction of 7% of VAP after implementation of SOD (but called ‘SDD’ in the paper). The use of ‘SDD’ was significantly associated with lower occurrence of VAP in a multivariate Cox regression model.

These findings are consistent with previous studies.

To my opinion, this study is interesting but there are some limitations that need to be discussed:

11)      The protocol used is not SDD but SOD! Selective oropharyngeal decontamination is less effective than SDD on clinical outcomes (Platinga et al., CMI, 2018), which can explain that no differences in length of hospitalization nor mortality rates have been found in this study.

22)       It is unclear if only the first episode of VAP has been taken into consideration? or if relapses or 2nd episode have been included in the analyses ?

33)      CMV reactivation is out of the scope of SOD/SDD field and I suggest that this part should be deleted.

44)      Of note, there is no evidence in literature for thinking that SOD/SDD can effectively prevent from pulmonary aspergillosis. Amphotericin B is included in the SOD/SDD for prevention of fungemia.

55)      Figure 1 provides no relevant information. Please add KM curves with VAP as event of interest with the two groups: with SOD and w/o SOD.

66)      Names of bacteria should be written in italics (in the text and in tables too)

77)      In table 1, patients with ‘SDD’ had a lower PaO2/FiO2 ratio, which reveals that patients of SDD group had a more severe presentation than the other group. This should be discussed.

88)      In-hospital mortality rates were dramatically high (over 50%). Do the authors have an explanation for this? Is the use of extracorporeal membrane oxygenation support was required in some cases?

99)      The conclusion (‘without any increase in the incidence of MDR bacteria’) is not supported by the methods and the results. There is no monitoring or systematic screening of MDR bacteria mentioned in the manuscript.

Author Response

Reviewer 2

  1. The protocol used is not SDD but SOD! Selective oropharyngeal decontamination is less effective than SDD on clinical outcomes (Platinga et al., CMI, 2018), which can explain that no differences in length of hospitalization nor mortality rates have been found in this study.

The decontamination preparation was applied in the oropharyngeal tract and the nasogastric tube (see methods). For this reason, we could not use the term SOD, but SDD without intravenous antibiotics. We fully agree that SDD without intravenous antibiotics resulted in being less effective than SDD plus intravenous antibiotics, and the issue has been mentioned in the discussion (see lines 213-220)

  1. It is unclear if only the first episode of VAP has been taken into consideration? or if relapses or 2ndepisode have been included in the analyses ?

Only the first VAP episode has been considered. We specified this point better in the methods.

  1. CMV reactivation is out of the scope of SOD/SDD field and I suggest that this part should be deleted.

In COVID19 patients, the CMV reactivation, as we recently published (Gatto et al, Intensive Care Med 2022), seems to be related (or caused) to bacterial VAP occurrence. As indicated by reviewer 1, this point could interest the reader.

  1. Of note, there is no evidence in literature for thinking that SOD/SDD can effectively prevent from pulmonary aspergillosis. Amphotericin B is included in the SOD/SDD for prevention of fungemia.

As for the CMV, we believe reporting data on COVID19 associated pulmonary aspergillosis may be interesting for the reader and the scientific community.

  1. Figure 1 provides no relevant information. Please add KM curves with VAP as event of interest with the two groups: with SOD and w/o SOD.

Modified as requested

  1. Names of bacteria should be written in italics (in the text and in tables too)

Modified as requested

  1. In table 1, patients with 'SDD' had a lower PaO2/FiO2 ratio, which reveals that patients of SDD group had a more severe presentation than the other group. This should be discussed.

We introduced a point in the limitations of the study.

  1. In-hospital mortality rates were dramatically high (over 50%). Do the authors have an explanation for this? Is the use of extracorporeal membrane oxygenation support was required in some cases?

In-hospital mortality of COVID19 patients requiring invasive mechanical ventilation, mostly after a non-invasive trial, with < 100 mmHg (mostly < 90 mmHg) is commonly reported to be > 50%, even with ECMO support in many of the reported experiences (Supady et al, Intensive Care Med 2022; Fanelli et al, Critical Care 2022; Lorusso et al, Lancet Resp Med 2022; Makhoul et al, Vaccines 2023). We want to specify that the intensive care and in-hospital mortalities of all the COVID19 patients (430 patients) admitted to our intensive care units, including patients requiring invasive and non-invasive mechanical ventilation, was 24 and 31% (in press Coloretti et al. Journal of Thoracic disease)

  1. The conclusion ('without any increase in the incidence of MDR bacteria') is not supported by the methods and the results. There is no monitoring or systematic screening of MDR bacteria mentioned in the manuscript.

We reported data on VAP caused by MDR in table 1 without differences in the SDD vs SDD treated patients. However, we agree that a lack of a systemic screening on MDR occurrence (e.g. rectal swab) is a study limitation. Therefore, we added this point to the limitations section of the study and changed the conclusions.

Reviewer 3 Report

Dear Authors

This paper examines the usefulness of selective digestive decontamination in preventing ventilator-associated pneumonia in COVID-19 patients. Although the introduction of SDD is interesting in that it reduces the occurrence of VAP, especially late VAP, there is no significant difference in survival between the VAP and noVAP groups. Conventionally, late VAP is thought to have a higher morbidity and mortality rate due to multidrug-resistant bacteria, so the results of this paper contradict this point. This needs to be reconsidered. Therefore, it was decided to reject.

Author Response

Reviewer 3

This paper examines the usefulness of selective digestive decontamination in preventing ventilator-associated pneumonia in COVID-19 patients. Although the introduction of SDD is interesting in that it reduces the occurrence of VAP, especially late VAP, there is no significant difference in survival between the VAP and noVAP groups. Conventionally, late VAP is thought to have a higher morbidity and mortality rate due to multidrug-resistant bacteria, so the results of this paper contradict this point. This needs to be reconsidered. Therefore, it was decided to reject

The relationship between VAP and mortality, as well as the effect of SDD in reducing mortality, have been and are still largely debated. However, several trials and metanalysis showed a significant effect of SDD in lowering the incidence of VAP with no or minimal effects on mortality. For instance, the recently published trial including 6000 patients (Myburgh JA, et al JAMA 2022) showed no change in hospital mortality. The subsequent metanalysis (Hammond NE et al  JAMA 2022) indicated only a minimal reduction in pooled estimated risk ratio (RR) for mortality for SDD compared with standard care (RR 0.91 -95% credible interval [CrI], 0.82-0.99; I2 = 33.9%; moderate certainty) with effect only in trials using SDD and intravenous antibiotics.

Round 2

Reviewer 2 Report

Fine with me, congrats to the authors for this improved version of the manuscript